# Three-Dimensional Foot Position Estimation Based on Footprint Shadow Image Processing and Deep Learning for Smart Trampoline Fitness System

**DOI:** 10.3390/s22186922

**Published:** 2022-09-13

**Authors:** Se-Kyung Park, Jun-Kyu Park, Hong-In Won, Seung-Hwan Choi, Chang-Hyun Kim, Suwoong Lee, Min Young Kim

**Affiliations:** 1Ansan R&D Campus, LG Innotek, Ansan 15588, Korea; 2Renewable Energy Solution Group, Korea Electric Power Research Institute (KEPRI), Naju 58277, Korea; 3Advanced Mechatronics Research Group, Daegyeong Division, Korea Institute of Industrial Technology, Daegu 42994, Korea; 4School of Electronics Engineering, Kyungpook National University, Daegu 41566, Korea; 5Research Center for Neurosurgical Robotic System, Kyungpook National University, Daegu 41566, Korea

**Keywords:** smart fitness, trampoline, 3D foot contact position estimation, wide-angle camera, footprint shadow, image processing, deep learning

## Abstract

In the wake of COVID-19, the digital fitness market combining health equipment and ICT technologies is experiencing unexpected high growth. A smart trampoline fitness system is a new representative home exercise equipment for muscle strengthening and rehabilitation exercises. Recognizing the motions of the user and evaluating user activity is critical for implementing its self-guided exercising system. This study aimed to estimate the three-dimensional positions of the user’s foot using deep learning-based image processing algorithms for footprint shadow images acquired from the system. The proposed system comprises a jumping fitness trampoline; an upward-looking camera with a wide-angle and fish-eye lens; and an embedded board to process deep learning algorithms. Compared with our previous approach, which suffered from a geometric calibration process, a camera calibration method for highly distorted images, and algorithmic sensitivity to environmental changes such as illumination conditions, the proposed deep learning algorithm utilizes end-to-end learning without calibration. The network is configured with a modified Fast-RCNN based on ResNet-50, where the region proposal network is modified to process location regression different from box regression. To verify the effectiveness and accuracy of the proposed algorithm, a series of experiments are performed using a prototype system with a robotic manipulator to handle a foot mockup. The three root mean square errors corresponding to X, Y, and Z directions were revealed to be 8.32, 15.14, and 4.05 mm, respectively. Thus, the system can be utilized for motion recognition and performance evaluation of jumping exercises.

## 1. Introduction

COVID-19 has had a significant impact on our daily life, especially our health [1,2,3]. In the wake of COVID-19, the digital fitness market combining health equipment and information and communication technology (ICT) is facing a significant growth opportunity; and the smart sports equipment market is also expected to grow in tandem. Smart sports equipment is digitally connected to sensors to monitor, track, and analyze the motion of athletes or exercises; and improve their performance [4,5]. Owing to the prolonged pandemic, personal exercises and home trainings have increased; vitalizing the smart sports equipment-related devices and service market. According to a report by Allied Market Research, the global smart sports equipment market size is expected to grow from approximately $7 billion in 2019 to $12 billion in 2026; with a CAGR of 7.9% over the same period [6]. 

Meanwhile, trampolines, which are familiar to men and women of all ages as playing devices for children, have recently gained popularity as exercise equipment; in addition, their effectiveness has been proven in muscle strengthening exercises [7,8,9] and rehabilitation exercises [10,11]. Recently, jumping fitness, which uses the trampoline to implement various movements of exercises indoors has gained popularity at home and abroad [12]. When trampolines are combined and linked with ICT, such as sensors, wired/wireless communication, and immersive contents, it is expected that people can easily perform and enjoy exercises at home to increase muscle strength and maintain balance. Moreover, it facilitates quantitative assessment of exercises and management of exercise data history. To implement smart sports equipment based on a trampoline, it is critical to recognize the user’s motions and evaluate the user activity during exercise.

As listed in Table 1, studies have been reported on systems that combine trampolines with ICT to recognize the motions of users. A previously conducted study classified motions on the trampoline by attaching inertial sensors to the arms, legs, thighs, and waist of trampoline players [13]. Studies have also been conducted on a system by integrating a high-speed camera and a three-axial accelerometer system; and attaching sensors to the hips to analyze the effect on motions based on the bouncing characteristics of the trampoline [14]; as well as a system that classified trampoline athletes’ motion using a camera [15]. These studies were primarily aimed at analyzing the motions of professional athletes. However, in all the aforementioned studies, the person jumping on the trampoline had to wear sensors; this was an inconvenience. Studies have been reported on a system that improves the motivation to exercise by connecting to an entertainment system via a distance sensor installed under the trampoline; and implements different movements, such as standing, walking, low jump, and high jump [16,17]; however, the system was limited in recognizing the user’s detailed motions. Research has been conducted on a system that connects a jumping game to a large trampoline using Microsoft Kinect; it improves jumping ability and motivates the user to exercise by exaggerating the movements on the screen [18]. Another study has been conducted on a system that recognizes head, arm, and leg movements using a motion-capture camera based on the linkage with virtual reality (VR) to determine the response level of the movements [19]. It increases the immersion and entertainment in the game by using a head mounted display (HMD) device. In these studies, user movements are primarily recognized using a camera installed outside the trampoline; however, it is impossible to recognize the jumping force or the contacting positions of the foot when jumping on the trampoline.

In a previous study, we proposed a smart trampoline fitness system (STFS) capable of recognizing the three-dimensional (3D) position of feet; more specifically, the feet position on the trampoline transverse plane and depth pressed by feet [20]. The prototype of STFS comprises a jumping fitness trampoline, ultra-wide-angle fish-eye camera, and an embedded board. Image processing-based algorithms for 3D foot contact position estimation were developed; and the positions were estimated using the binary image of the foot shadow obtained by extracting the binary masks of the trampoline and foot. Although the system cannot recognize the movement of the user’s entire body, the 3D foot contact position estimation algorithms have several technological benefits. First, the users of the system are not required to wear sensors such as those used in [13,14]; therefore, they can exercise freely. Moreover, the user’s motions can be recognized in more detail compared to [16,17]. Moreover, the system is relatively simple and can be constructed at a lower cost compared to [14,18,19]. Finally, the algorithms can detect the intensity of jumping as well as foot position on a trampoline as compared to [15,18,19]. As a result of estimating the 3D position of the foot using an image processing-based algorithm in the previous study [20], we encountered problems with linear coefficient extraction; an ultra-wide-angle correction coefficient; and those that are significantly affected by surrounding objects and environments.

In order to solve these problems, we propose an algorithm using deep learning; moreover, the objective of this paper is to estimate the 3D position of the foot using the footprint shadow image processing and deep learning for STFS. The details of the methodology for the proposed approach are described in this paper. The remainder of this study is organized as follows. Section 2 introduces the method for the 3D position of the foot using the footprint shadow image processing and deep learning. Section 3 describes the results of experiments on the deep learning-based algorithm. The proposed methodology for the 3D foot position estimation is discussed in Section 4. Finally, the conclusions of this study are presented in Section 5.

## 2. Methods 

### 2.1. Overview of a STFS and Experimental Environment

Figure 1 displays a prototype of a STFS. It comprises a jumping fitness trampoline (J6H130 FLEXI, W × D × H = 1360 × 1360 × 285 (mm), maximum load of 130 kg, Jumping Inc., Prague, Czech Republic); an embedded processor for real-time image processing and deep learning (Jetson AGX Xavier, 512-core Volta GPU with Tensor Cores, 8-core ARM v8.2 64-bit CPU, 8 MB L2 + 4 MB L3, 32 GB 256-Bit LPDDR4x|137 GB/s, 7-way VLIW Vision Processor, Nvidia, Santa Clara, CA, USA); and a camera module (UC-626 rev. B, maximum 8 mega pixels: 3264 × 2448 pixels, maximum 30 fps at 1920 × 1080 pixels, Arducam, Hong Kong) with a 220° wide-angle fish eye lens. The part of the trampoline where the user’s feet makes contact is composed of a translucent material; thus, the shadow of the foot in contact is clearly visible when viewed from below. The camera module was attached to the bottom of the trampoline to capture the shadow of the foot. The extracted shadows were used to estimate the contact position on the trampoline transverse plane (X–Y) and the depth pressed by the foot (Z).

Figure 2 shows the experimental environment for collecting learning and test data for 3D foot position estimation. First, the foot-shaped contact jigs were manufactured according to human foot sizes ranging from 210–280 mm in 10 mm increments. The jig of each size was changed and attached to the end-effector of the robot manipulator (iiwa 14 R820, payload of 14 kg, precision of ±0.15 mm, reach of 820 mm, KUKA, Augsburg, Germany). 

### 2.2. Network Structure for 3D Foot Position Estimation Based on Deep Learning

The correct detection coordinate labels are required to understand the existing image detection process. However, the use of custom data rather than open data can limit the labeling work. Therefore, the model was constructed using the obtainable image data and coordinate data of the manipulator. Since the image data, foot size, and real contact coordinates of the manipulator X,  Y, Z are known, the image and foot size were used as input data; and the coordinates of the manipulator X, Y, Z  were learned as correct data—that is, the model was devised such that the estimated coordinates $\hat{X}$, $\hat{Y}$, $\hat{Z}$ formed the output. Accordingly, it was determined that the $\hat{Z}$ value estimated based on the foot size could vary; consequently, the foot size could be added to the input data. Faster R-CNN [21] using a region proposal network (RPN) is unsuitable as a real-time model; however, owing to the lack of labels, the learning direction of the model needed to be conceived and verified. Consequently, the 3D foot position estimation based on a deep learning (3DFPE-DL) algorithm was implemented using the corresponding model. Figure 3 illustrates the overall flow of the 3DFPE-DL.

The image entered as the input in Figure 3 is shown in Figure 4a. Following conversion from RGB to grayscale considering its use in the embedded processors, only the required area was cropped (from 640 × 480 to 480 × 480 pixels). The image was then min-max normalized to proceed with the learning process following the image preprocessing of the grayscale image (Figure 4), as depicted in Figure 4b.

The feature extractor selected for the base network was ResNet [22], which has a repeated shape of the residual block (using a bottleneck architecture). The network preserves existing learning information and performs additional learning; in addition, it can be characterized by solving the vanishing gradient problem in which the existing learning information is forgotten as the number of layers increases by connecting the information learned in the previous layer. ResNet-50 comprises 50 layers, as presented in Table 2. The output_size is downsampled as it passes through the convolutional layer in the convolution block, as shown in Figure 5a; and learns the identity block, as shown in Figure 5b, which is in the shape of a typical residual block.

In the existing Faster R-CNN, the feature map extracted through the feature extractor is subjected to classification and box regression using the RPN. However, in the 3DFPE-DL, location regression is performed instead of the box regression of the RPN; as illustrated in Figure 6.

When performing the RPN, as shown in Figure 6, the previously extracted feature map and foot size, as an input value, are merged using the concatenate step and entered as an input to the RPN. The foot coordinates $\hat{X}$, $\hat{Y}$, $\hat{Z}$ can be estimated through the corresponding network.

Since the correct data and extraction results are coordinates, the loss function between the correct answer coordinates X,  Y, Z  and estimated $\hat{X}$, $\hat{Y}$, $\hat{Z}$ was selected as the mean square error (MSE); and learning was conducted.

### 2.3. Experimental Method for Data Acquisition

Considering the operating range and diameter of the manipulator jig, the two-dimensional contact position of the learning data was selected. The center of the trampoline was set as the origin; and the position of the two-dimensional reference was selected within a movable distance of 0, 50, 100, 150, 200, and 250 mm from the origin. The 48 positions considered in the experiment are shown in Figure 7a. Measurements were performed in units of 10 mm each; from 0 to 70 mm perpendicular depth at the blue point position and from 0 to 100 mm at the green point position. In total, 403 3D positions were selected, data were collected by rotating from 20° to 340° in steps of 10° at each position, and a total of 106,392 measurements were performed (13,299 for each size).

Furthermore, the contact position of the manipulator is represented as a two-dimensional area, as shown in Figure 7b, to collect the validation and test data. The green area in Figure 7b represents an area of distance within 180 mm from the origin and can be measured from 0 to 100 mm of vertical depth. The blue area corresponds to an area of distance within 250 mm from the origin, excluding the green area; and can be measured from 0 to 70 mm of vertical depth. The coordinates and rotation angles of the manipulator were randomly selected within the corresponding range; and a total of 24,000 test measurements were performed with a total of 4000 confirmation measurements being collected (3000 and 500 for each size, respectively).

### 2.4. Statistical Method for System Evaluation

The distance error between the actual position acquired by the manipulator and estimated position computed by 3DFPE-DL was considered for the proposed system evaluation. For this purpose, the root mean square error (RMSE) expressed in Equation (1) was used as a statistical method. RMSE is a standard method to measure the error of a model in predicting quantitative data. Here, $P_{i}$ indicates the actual contact position, X, Y, Z; and ${\hat{P}}_{i}\;$ indicates the estimated contact position, $\hat{X}$, $\hat{Y}$, $\hat{Z}$.
(1)$$\mathrm{RMSE}=\sqrt{\frac{\sum_{i=1}^{N}{\left(P_{i}-\hat{P_{i}}\right)}^{2}}{N}}$$

## 3. Results

Table 3 lists the standard error between the 3D point and each coordinate axis. Figure 8 presents the results of Table 3 as a graph. It can be confirmed that while the difference between the minimum and maximum values is large due to some measurement noise, the average values show that the distance error is less than 12 mm. Relatively higher error values of x and y coordinates than that of z coordinate apparently result from the lens distortion effect of the wide view camera. Moreover, the results indicate that the *y*-axis distortion is greater that of the *x*-axis.

Table 4 and Figure 9 present the results obtained by analyzing the average error based on the area from the origin. The green area in Figure 7b includes distance ranges of 50, 100, and 150 mm; and the blue area includes a distance of 200 and 250 mm. From the table, it is evident that the largest error occurs at 250 mm in the range including the blue area; with an error at 200 mm greater than the average error occurring in the green area. In other words, the closer to the boundary of the trampoline, the greater the increase in error.

Table 5 and Figure 10 present the average error based on the area of distance from the origin to a vertical depth. The trends indicated in Table 4 can also be observed in Table 5; that is, the error is smaller based on a depth of 70 mm. It is evident that the depth of 0 to 70 mm includes both the green and blue areas of Figure 7b. However, that of 70 to 100 mm includes only the green areas; indicating the error including the blue area is larger, as confirmed in Table 4.

Unlike the experimental environment of [20], here, the data were randomly extracted for a certain area rather than a certain location. Consequently, an error could be confirmed by dividing the area into a range. During analysis of the cause of errors in the blue part of Figure 7b, data—which included interference of the shade of the manipulator with the outer periphery of the trampoline—were generated; as displayed in Figure 11. It should be mentioned that it was difficult to organize all the interference data with random coordinates and rotation angles within a certain area.

It can be stated that the results did not deviate from the purpose of this experiment, considering that there was versatility for each size compared to the previous experiment; the locations of the learning data and test data differed; and the surrounding objects changed when collecting the data. Figure 12 shows the 3D foot position estimated when an experimental subject jumped step-by-step on the trampoline. 

## 4. Discussion

In this study, as a basic experiment, the 3D position of one foot was estimated by acquiring one foot shadow data using a robot and jig. To estimate the contact positions of two feet, the experimental environment comprising a robot manipulator and foot contact jig should be improved; and then, the learning and verification processes of the 3DFPE-DL algorithm are required to proceed. However, it is also possible to review the estimation method using the 3DFPE-DL algorithm by integrating with the image processing-based algorithm in [20]. Nevertheless, the results of this study can be considered as a function of supplementing the existing 3D foot position estimation algorithm based on image processing.

The experimental results revealed that the 3DFPE-DL algorithm could solve problems with a linear coefficient extraction; with an ultra-wide-angle correction coefficient; and those which are significantly affected by surrounding objects and environments. However, the estimation of the foot contact position in the Z-direction compared to that in the X-Y direction was relatively inaccurate; a similar trend was reported in a previous study [20]. A possible reason can be the use of distorted images due to the use of a wide-angle lens for learning. Considering the linkage with the content, the foot contact position in the Z direction is used to determine the relative strength or contact; therefore, it would be sufficient to produce a relative numerical value from a practical perspective.

As an example, a STFS linked with game contents, as shown in Figure 13a, can be made based on the result of this study, by designating a range of pressing areas (Figure 13b). This case may not require high accuracy for the 3D foot position estimation. However, it is also possible to link content that analyzes the motions of professional athletes. In this case, a relatively high accuracy of the foot contact position estimation may be required because the location of the landing point on the trampoline is crucial [23,24,25]. Therefore, for general usage, such as home exercise equipment, as shown in Figure 13c, the results derived from this study may be sufficient. However, when the results are used for professional use, additional performance improvement may be necessary.

## 5. Conclusions

In this study, we proposed a 3DFPE-DL for STFS. A system prototype of the STFS was developed using a trampoline for jumping fitness, an embedded processor capable of deep learning, and an ultra-wide-angle fisheye camera. Experiments for collecting learning and test data for 3D foot-position estimation were conducted in an environment that consisted of foot-shaped contact jigs and a robot manipulator. The 3DFPE-DL algorithm was configured with a ResNet-50 based Fast-RCNN, where the RPN was changed to suit the purpose of 3D foot position extraction. As an experimental result, RMSE values corresponding to X, Y, and Z directions of 8.32, 15.14, and 4.05 mm, respectively, were derived through the 3DFPE-DL algorithm.

As a future work, the 3DFPE-DL algorithm should be stabilized and improved to a 1-stage model; referring to the problems considered in the conceptual stage. Moreover, the 3DFPE-DL algorithm could be applied to low-cost embedded processors by simplifying the algorithm. The results for foot position estimation in the Z direction can be linked to the trampoline energy consumption prediction mathematical model [26], and it is expected to present exercising effects to users of the STFS.

## Figures and Tables

**Figure 1 sensors-22-06922-f001:**
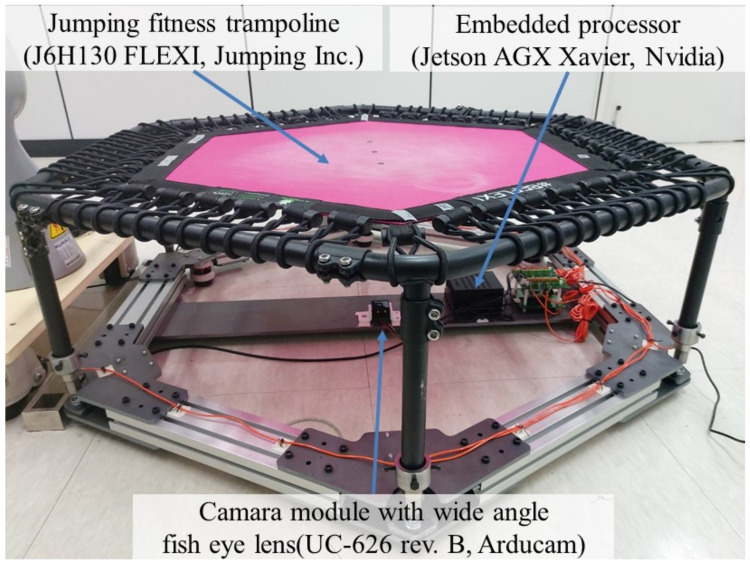
Prototype of a smart trampoline fitness system (STFS).

**Figure 2 sensors-22-06922-f002:**
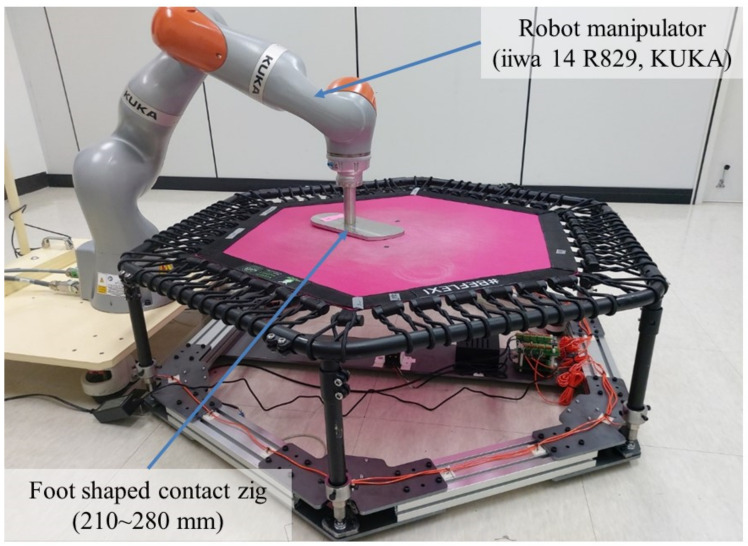
Experimental environment for collecting the learning and test data.

**Figure 3 sensors-22-06922-f003:**
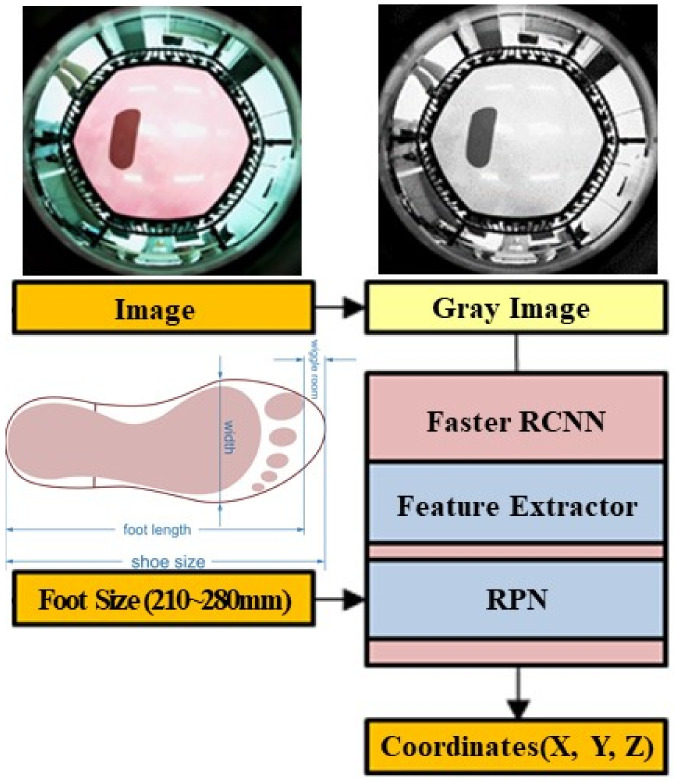
Flow of the 3DFPE-DL algorithm.

**Figure 4 sensors-22-06922-f004:**
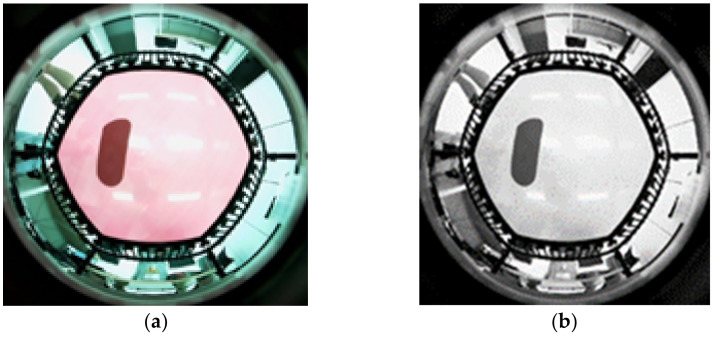
The input image. (**a**) Source image. (**b**) Preprocessing input image.

**Figure 5 sensors-22-06922-f005:**
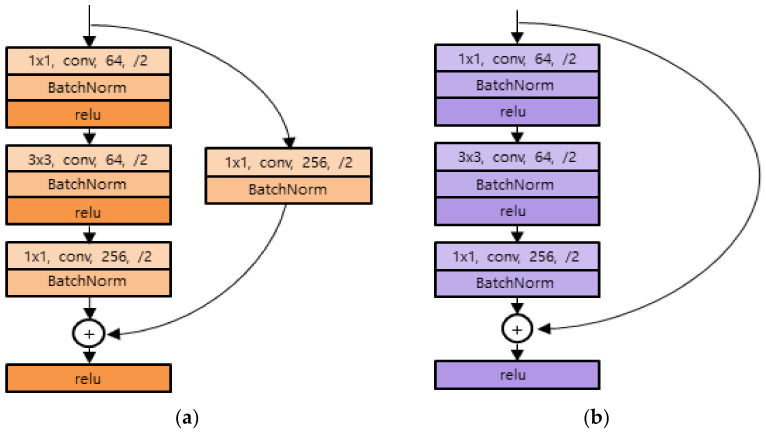
ResNet-50 structure and the residual block. (**a**) Convolution_block. (**b**) Identity_block.

**Figure 6 sensors-22-06922-f006:**
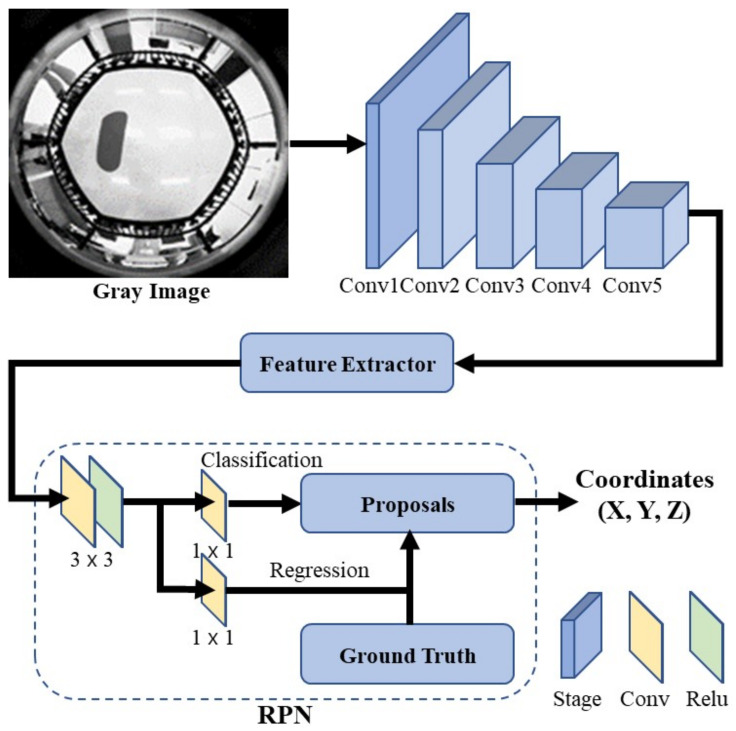
RPN in a Faster R-CNN for a 3DFPE-DL.

**Figure 7 sensors-22-06922-f007:**
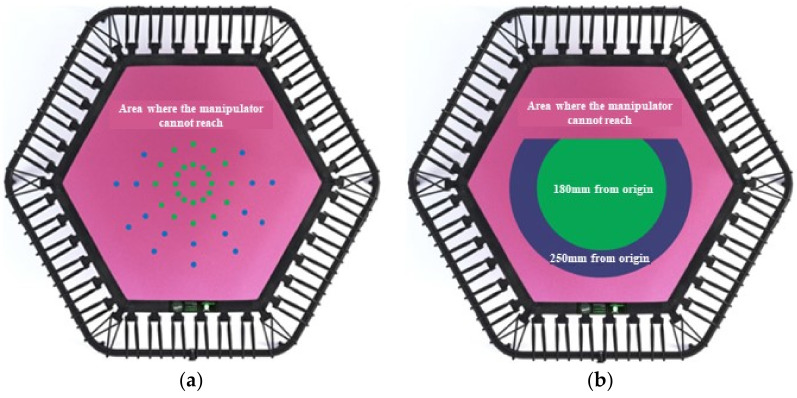
2D contact positions of the manipulator on the trampoline. (**a**) Contact positions of the learning data. (**b**) Contact positions of the validation and test data.

**Figure 8 sensors-22-06922-f008:**
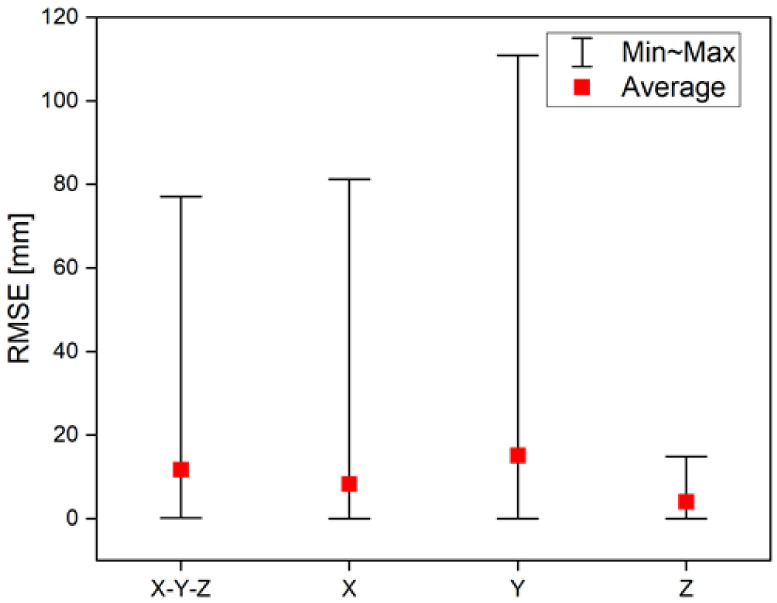
RMSE graph of the X-Y-Z estimation position.

**Figure 9 sensors-22-06922-f009:**
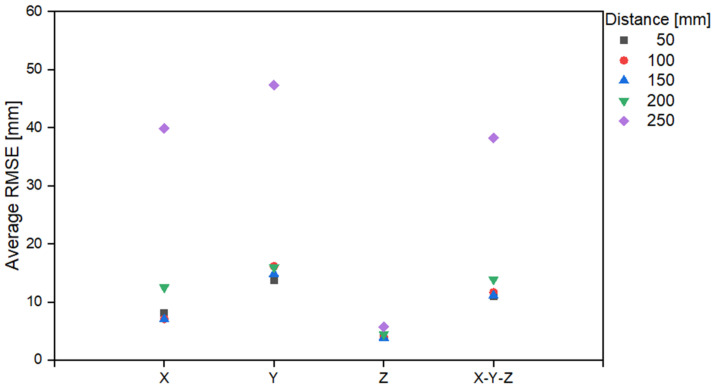
RMSE graph based on the X-Y distance.

**Figure 10 sensors-22-06922-f010:**
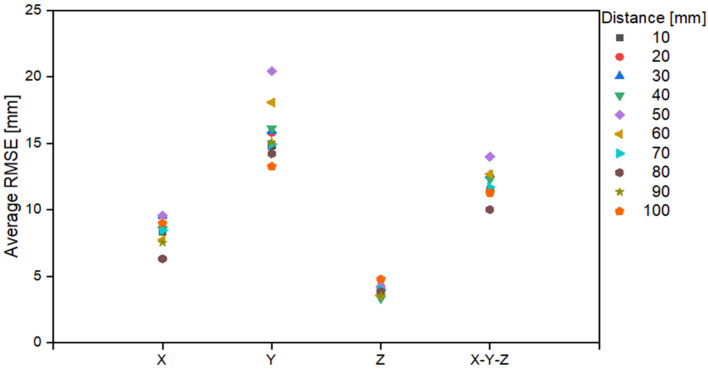
RMSE graph based on the Z distance.

**Figure 11 sensors-22-06922-f011:**
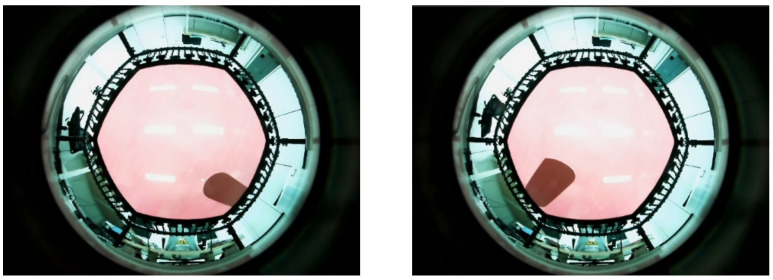
Interference image between the manipulator and the trampoline.

**Figure 12 sensors-22-06922-f012:**
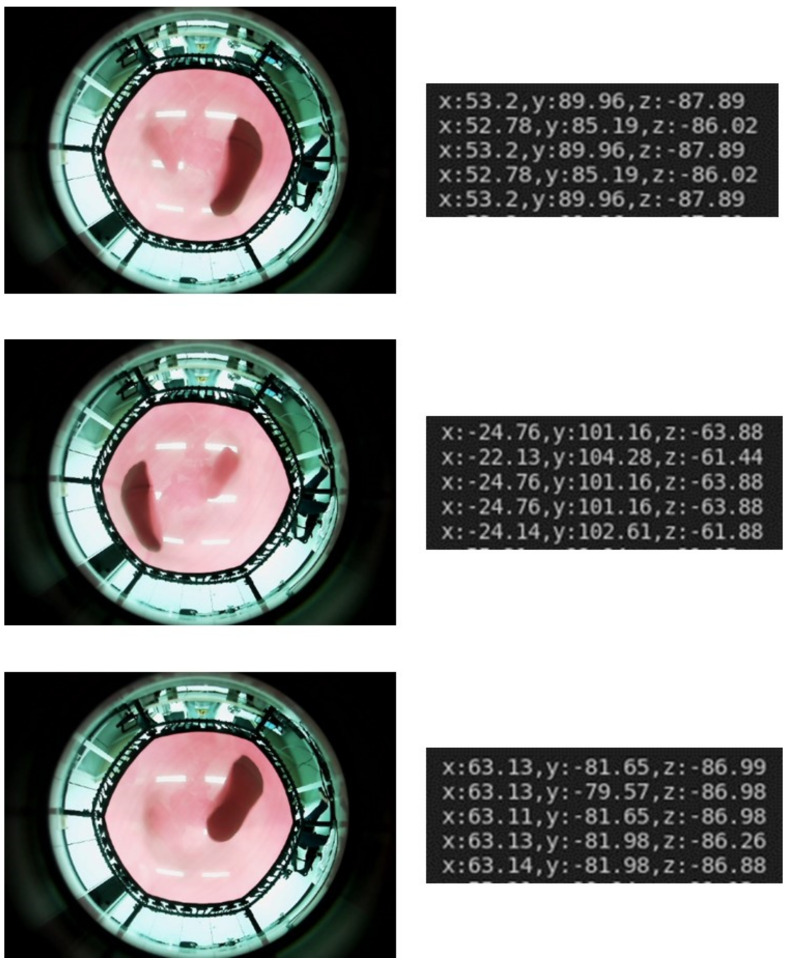
3D foot position of the experimental subject estimated by the 3DFPE-DL algorithm (**left**: images acquired by the camera module; **right**: numerical values of the 3D foot position estimated in 0.1 s increments).

**Figure 13 sensors-22-06922-f013:**
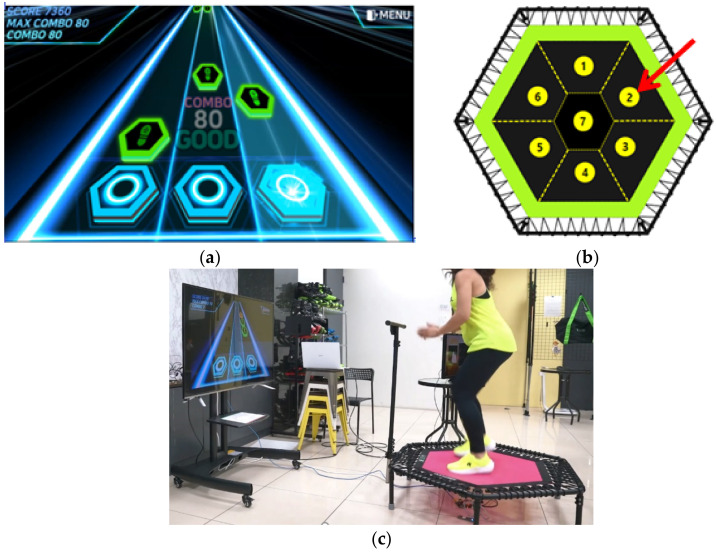
Example of a STFS linked with game contents based on the results of this study. (**a**) Example of game contents (rhythm game) for a STFS. (**b**) Example of a range of pressing areas. (**c**) Usage example of a STFS for home exercise equipment.

**Table 1 sensors-22-06922-t001:** Pros and cons of a trampoline system using sensors.

System Type	Pros/Cons	
Classification of trampoline athletes’ motion using inertial sensors [10].	Pros	Allows the correct classification of athletes’ movements.
	Cons	Requires multiple sensors to be installed. Difficult to use for the public as it is designed for athletes.
Characteristic analysis system of trampoline bounce using a high-speed camera and 3-axis accelerometer [11].	Pros	Effect of bounce characteristics on emotional response can be determined.
	Cons	No association with exercise and games. Requires a sensor to be attached to the hip.
Classification of trampoline athletes’ motion using a camera [12].	Pros	Allows the filtering and classification of players’ poses.
	Cons	Requires secure spacing between the camera and trampoline. Difficult to use for the public as it is designed for athletes.
Analysis of a status and content integrated system using distance sensors [13,14].	Pros	Increases motivation for exercise through content integration. Available to the public.
	Cons	Can only classify walking, low jumping, and high jumping.
Jumping game integrated system using Kinect [15].	Pros	Increases jumping power through content integration. Available to the public.
	Cons	Requires secure spacing between Kinect and the trampoline.
VR integrated gaming system using a motion-capture camera and HMD [16].	Pros	Improves the immersion and enjoyment of games.
	Cons	Requires complex safety equipment. Requires arm and leg sensors and headgear.

**Table 2 sensors-22-06922-t002:** ResNet-50 structure in this study.

Layer_Name	Output_Size	ResNet-50 Layer
Conv1	112×112	7×7, 64, stride 2
Conv2_x	56×56	3×3, max pool, stride 2
		$\left[\begin{array}{c} \begin{array}{cc} 1\times 1 & 64\\ 3\times 3 & 64 \end{array}\\ \;1\times 1\;\;\;256 \end{array}\;\right]\times 3$
Conv3_x	28×28	$\left[\begin{array}{c} \begin{array}{cc} 1\times 1 & 128\\ 3\times 3 & 128 \end{array}\\ 1\times 1\;\;\;\;\;512 \end{array}\;\right]\times 4$
Conv4_x	14×14	$\left[\begin{array}{c} \begin{array}{cc} 1\times 1 & 256\\ 3\times 3 & 256 \end{array}\\ \;1\times 1\;\;\;1024 \end{array}\;\right]\times 6$
Conv5_x	7×7	$\left[\begin{array}{c} \begin{array}{cc} 1\times 1 & 512\\ 3\times 3 & 512 \end{array}\\ \;1\times 1\;\;\;2048 \end{array}\;\right]\times 3$
	1×1	Average pool, 1000-d fc, SoftMax
FLOPs		$3.8\times 10^{9}\;$

**Table 3 sensors-22-06922-t003:** RMSE of the X-Y-Z estimation position.

	RMSE [mm]		
	Min	Max	Average
X-Y-Z	0.2	77.1	11.7
X	0.0	81.2	8.3
Y	0.0	110.9	15.1
Z	0.0	14.9	4.1

**Table 4 sensors-22-06922-t004:** RMSE based on the X-Y distance.

Area Color	Distance Range [mm]	RMSE [mm]			
		X	Y	Z	X-Y-Z
Green	50	8.1	13.7	4.1	11.0
	100	7.1	16.2	4.0	11.7
	150	7.0	14.8	3.9	11.1
Blue	200	12.5	16.0	4.5	13.9
	250	39.9	47.4	5.7	38.2

**Table 5 sensors-22-06922-t005:** RMSE corresponding to the Z distance.

Depth Range [mm]	RMSE [mm]			
	X	Y	Z	X-Y-Z
10	8.4	14.8	4.0	11.5
20	9.0	15.8	4.2	12.5
30	9.6	15.9	4.1	12.6
40	8.5	16.1	3.4	12.3
50	9.6	20.4	4.2	14.0
60	7.8	18.1	3.6	12.7
70	8.5	14.9	4.0	11.7
80	6.3	14.2	3.9	10.0
90	7.5	15.1	3.6	11.4
100	9.0	13.3	4.8	11.3

## Data Availability

Not applicable.

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
