# Peer review of "Three-Dimensional Foot Position Estimation Based on Footprint Shadow Image Processing and Deep Learning for Smart Trampoline Fitness System"

_sensors, 2022, doi:10.3390/s22186922_

Round 1

Reviewer 1 Report

Dear Authors,

Congratulations for the work done. It is a well-developed research and of interest to the potential readers of this journal. However, the manuscript has some methodological limitations that should be resolved before its possible publication in this Journal.

ABSTRACT: The authors should include a sentence that contextualizes the research by addressing its background. The Results section is not correctly represented in this section.
In addition, abbreviations should be avoided if possible. The verb tenses used are not correct.

INTRODUCTION: The COVID-19 pandemic is well contextualized but the health impact of the pandemic should be taken into account (doi: 10.3390/ijerph18010031).
The contextualization of human motion analysis through sensors should take into account relevant research in this field on accelerometers and other inertial systems.

METHODS: It is not correct to title this section with an added nickname.
Authors should include a final subsection explaining the statistical techniques applied (including THE STATISTICAL POWER OF THE RESEARCH).

RESULTS: Zeros as the last decimal place should be deleted. Authors should reconsider the need to transmit results with two decimal places (do they really provide additional or superior information to that transmitted by a single decimal place?).

DISCUSSION: This section could also benefit from the inclusion of more and new bibliographic references to support the text.

Kind regards.

Author Response

The authors really appreciate Reviewer 1’s comments regarding the manuscript.

As mentioned by Reviewer 1, the manuscript contains some methodological limitations that needs to be resolved. Accordingly, some sentences in the manuscript have been revised based on his/her comments as follows (the edited parts of the manuscript are marked in red).

ABSTRACT: The authors agree that the abstract was missing the background of this study and added a sentence that contextualizes the research by addressing the background in Lines 17-21. As the evaluation of the test results were omitted in the abstract, related sentences were added in Lines 33-34. All abbreviations were deleted and the verb tenses used are revised correctly in the ABSTRACT.

INTRODUCTION: As mentioned by Reviewer 1, the health impact of the pandemic should be considered. The authors added a related sentence in Line 38 and references [1–3] including a related study as suggested by Reviewer 1. Moreover, a sentence was added in Lines 60-62 to cite studies on motion recognition in various trampoline systems to improve the context of the INTRODUCTION.

METHODS: The authors agree that it is not correct to title this section with an added nickname. Accordingly, the nickname was deleted. Additionally, a subsection explaining the applied statistical techniques was added as Lines 226-233.

RESULTS: The main view area of the camera looking the trampoline is about 1360x1360mm in x and y directions, and the corresponding imaging area is about 700x700 pixel. It means the pixel resolution is about 1.9mm. If a tenth is achievable as general machine vision, a single decimal place is useful. Therefore, we changed the results into ones with a single decimal point in all Tables.

And some zeros were deleted.

DISCUSSION: The authors revised sentences in Line 307-315, added references [23-25] to discuss the results of this study, and divided the equipment into general use and professional use. In particular, for general use, such as home exercise equipment, high accuracy for the 3D foot position estimation may not be required and the results derived from this study may be sufficient. However, relatively high accuracy of the foot contact position estimation may be required when the results are used for professional use that analyzes the motions of professional athletes, primarily because the location of the landing point on the trampoline is crucial.

Added references

  1. World Health Organization. https://www.who.int/campaigns/connecting-the-world-to-combat-coronavirus/healthyathome (accessed on 25 Aug. 2022)
  2. Sarah Alonzi, et al. “The psychological impact of preexisting mental and physical health conditions during the COVID-19 pandemic” Psychological Trauma: Theory, Research, Practice, and Policy 12. S1 (2020)
  3. Óscar Rodríguez-Nogueira, et al. “Musculoskeletal Pain and Teleworking in Times of the COVID-19: Analysis of the Impact on the Workers at Two Spanish Universities”, International Journal of Environmental Research and Public Health 18.1 (2020)
  4. Jens Natrup, et al. “Gaze behavior of trampoline gymnasts during a back tuck somersault”, Human Movement Science 70.3 (2020).
  5. “Trampoline & Tumbling Code Of Points”, AAU Trampoline and Tumbling (2021)

25. BBC. Available online : https://www.bbc.co.uk/bitesize/guides/zp99j6f/revision/3 (accessed on 25 Aug. 2022)

Reviewer 2 Report

The current study succeeded in developing the foot position estimation system from foot-shadow images utilising deep learning in the trampoline equipment. Basically, deep learning approach seemed to be adequate and relatively low measurement errors were achieved. The fundamental concerns, however, remain as follows.

1. Introduction does not provide sufficient justification about why the proposed technology is important. Why should the three-dimensional feet positions be monitored? Does it help trampoline exercise in any way? This is one of the fundamental questions in the current manuscript because the introduction section did not justified why foot position detection should be based on shadow recognition and how such technology practically benefits.

2. Line 52: ‘grow the height’ Is it really so? Please explain a bit more in detail.

3. Line 60-64: ‘First, studies have been conducted …motion using a camera [12].’ This sentence is too long. Break it up into a few sentences.

4. Line 84-95: ‘In a previous study…by surrounding objects and environments.’ Why were feet positions and depth important to be recorded? What are practical applications? Why was the proposed system better than others (e.g. cost effectiveness, accuracy)?

5. Table 5: Compared to x coordinates, y coordinates seemed to have higher measurement errors. In the transverse plane, are there any potential reasons why this happened?

6. Figure 16 (b): This illustration has not been really referred within the main text. However, it would not require foot shadow and deep learning if the purpose is to recognise foot landing in these 7 areas. To justify the current system, more detailed foot position recognition should be expected.

Author Response

The authors really appreciate Reviewer 2’s comments regarding the manuscript. Accordingly, the authors have revised content in the manuscript based on his/her comments (the edited parts of the manuscript are marked in red).

  1. To implement smart sports equipment based on a trampoline, it is critical to recognize the user’s motions and evaluate the user activity during exercise because the result of the motion recognition can be linked with software content, quantitative assessment of exercises, and management of exercise data history. The authors have added related sentences in Lines 58-60 and 76–80.

  1. The authors agree that the phrase ‘grow the height’ is inappropriate. Accordingly, it was deleted.

  1. The authors agree that the sentence mentioned by Reviewer 2 is too long and have divided it into two sentences expressed in Line 76-82.

  1. Although the system cannot recognize the movement of the user's entire body, the 3D foot contact position estimation algorithms have several technological benefits. For instance, First, users of the system are not required to wear sensors such as those used in [13, 14], so they can exercise freely. Besides, the user’s motions can be recognized in more detail compared to [16, 17]. Moreover, the system is relatively simple and can be constructed at a lower cost compared to [14, 18, 19]. Finally, the algorithms can detect the intensity of jumping as well as foot position on a trampoline compared to [15, 18, 19]. The authors have added related sentence in Line 105-112.

  1. Relatively higher error values of x and y coordinates than that of z coordinate seems to result from the lens distortion effect of the wide view camera. Moreover, the results indicate that the y-axis distortion is greater that of the x-axis. The authors have added related sentence in Line 240-43.

  1. As you can see in Table 3-5, the similar error behaviors are observed. As one of main causes, we think the image distortion due to the fisheye lens characteristics of the wide view camera. So, the following statement was included in the front of Table 3 (Line 240-43).

  1. The authors revised the sentences in Line 307-314 and added references [23-25] to discuss the results of this study, divided into general use and professional use. Especially for general use, such as home exercise equipment, high accuracy for the 3D foot position estimation may not be required and the results derived from this study may be sufficient. However, when the results are used for professional use that analyzes the motions of professional athlete, relatively high accuracy of the foot contact position estimation may be required because the location of the landing point on the trampoline is crucial.

Added references

  1. World Health Organization. https://www.who.int/campaigns/connecting-the-world-to-combat-coronavirus/healthyathome (accessed on 25 Aug. 2022)
  2. Sarah Alonzi, et al. “The psychological impact of preexisting mental and physical health conditions during the COVID-19 pandemic” Psychological Trauma: Theory, Research, Practice, and Policy 12. S1 (2020)
  3. Óscar Rodríguez-Nogueira, et al. “Musculoskeletal Pain and Teleworking in Times of the COVID-19: Analysis of the Impact on the Workers at Two Spanish Universities”, International Journal of Environmental Research and Public Health 18.1 (2020)
  4. Jens Natrup, et al. “Gaze behavior of trampoline gymnasts during a back tuck somersault”, Human Movement Science 70.3 (2020).
  5. “Trampoline & Tumbling Code Of Points”, AAU Trampoline and Tumbling (2021)
  6. BBC. Available online : https://www.bbc.co.uk/bitesize/guides/zp99j6f/revision/3 (accessed on 25 Aug. 2022)

Round 2

Reviewer 1 Report

Dear Authors,

Congratulations for the work done.

The corrections made have improved the methodological limitations and formal errors present in the previous version of the manuscript.

For this reason, I consider that it can now be accepted for publication in this Journal.

Kind regards.

Reviewer 2 Report

The authors have addressed my comments.

It will be even better if the authors can properly describe future applications in addition to the current trampoline case.

Otherwise, the application looks somewhat limited.

Nevertheless, I believe the technology and approach are interesting.